# Modeling PCDH19-CE: From 2D Stem Cell Model to 3D Brain Organoids

**DOI:** 10.3390/ijms23073506

**Published:** 2022-03-23

**Authors:** Rossella Borghi, Valentina Magliocca, Marina Trivisano, Nicola Specchio, Marco Tartaglia, Enrico Bertini, Claudia Compagnucci

**Affiliations:** 1Genetics and Rare Diseases Research Division, Bambino Gesù Children’s Research Hospital, IRCCS, 00165 Rome, Italy; rossella.borghi@opbg.net (R.B.); valentina.magliocca@opbg.net (V.M.); marco.tartaglia@opbg.net (M.T.); enricosilvio.bertini@opbg.net (E.B.); 2Department of Neurosciences, Rare and Complex Epilepsy Unit, Division of Neurology, Bambino Gesù Children’s Hospital, IRCCS, Full Member of European Reference Network EpiCARE, 00165 Rome, Italy; marina.trivisano@opbg.net (M.T.); nicola.specchio@opbg.net (N.S.)

**Keywords:** PCDH19, iPSCs, neurons, brain organoids, neurogenesis, disease model

## Abstract

PCDH19 clustering epilepsy (PCDH19-CE) is a genetic disease characterized by a heterogeneous phenotypic spectrum ranging from focal epilepsy with rare seizures and normal cognitive development to severe drug-resistant epilepsy associated with intellectual disability and autism. Unfortunately, little is known about the pathogenic mechanism underlying this disease and an effective treatment is lacking. Studies with zebrafish and murine models have provided insights on the function of PCDH19 during brain development and how its altered function causes the disease, but these models fail to reproduce the human phenotype. Induced pluripotent stem cell (iPSC) technology has provided a complementary experimental approach for investigating the pathogenic mechanisms implicated in PCDH19-CE during neurogenesis and studying the pathology in a more physiological three-dimensional (3D) environment through the development of brain organoids. We report on recent progress in the development of human brain organoids with a particular focus on how this 3D model may shed light on the pathomechanisms implicated in PCDH19-CE.

## 1. Introduction

PCDH19 clustering epilepsy (PCDH19-CE; MIM 300088) is a rare form of drug-resistant epilepsy caused by mutations or partial deletions of the PCDH19 gene [1]. It is characterized by fever-induced seizures, which arise in early childhood and that usually occur in clusters [1,2,3]. It is estimated that about 70% of affected individuals show intellectual disability (ID) of varying degrees [4,5], with a proportion of them displaying behavioral problems and autistic features [6,7,8,9,10,11]. In adolescents and adults, depression, bipolar behavior, schizophrenia, psychosis and other mental illnesses have also been described [6,12]. Other recurring symptoms include fine and gross motor delays, language delay, sensory integration difficulties, sleep problems, low motor tone, and constipation [1,3,6]. The frequency of seizures decreases with age and some patients became seizure-free in adolescence or adulthood; nevertheless, ID generally persists after seizure remission [13]. Patients with severe symptoms often require assistance and speech, motor and psychological therapy, both for the severity of epilepsy and their cognitive deficits, and their behavioral disorders.

Recently, brain abnormalities, including mild microcephaly [14], cortical dysplasia [15,16,17,18], and bilateral reductions of local gyrification index in limbic cortical areas [19] have been reported for PCDH19-CE, although brain imaging with nuclear magnetic resonance at onset of the seizures is typically described as normal.

To study the pathogenetic mechanism implicated in PCDH19-CE, work was originally performed using animals (i.e., zebrafish and mice) as model systems. These models, however, failed to fully recapitulate the phenotype and brain malformations characterizing the disease [16,17,20,21,22,23], since the development and the anatomy of human and animal brains show important differences. To overcome these issues and the lack of an informative experimental model that faithfully recapitulates the clinical phenotype, other experimental models have subsequently been considered. Among these, human induced pluripotent stem cell (iPSC) technology avoids the influence of the mouse genetic background to develop a species-specific model of the disease, in principle. iPSCs are reprogrammed cells with pluripotent behavior that can be obtained from patients’ somatic cells [24], and are able to differentiate into multiple cell types, including neuronal cells [25]. Being a bidimensional (2D) model, however, they are not able to recreate the cellular stratification of the brain. This limit has been overcome by the recent development of 3D organoids, which more faithfully recreate the brain structure and the interaction of different neuronal cell types with the genetic background of the donor [26,27]. Thanks to these characteristics, brain organoids represent a unique opportunity to model developmental disorders such as PCDH19-CE.

## 2. PCDH19-CE: Genetics

The PCDH19 gene is localized on the long arm of chromosome X (Xq22.3). Unlike “classical” X-linked disorders, PCDH19-CE mainly affects females and is characterized by an unusual pattern of inheritance. Recessive X-linked disorders typically affect males, with heterozygous females expressing mitigated clinical features in a subset of these diseases. On the other hand, in dominant X-linked disorders, females are typically affected, with some of these conditions being embryonic lethal for males. In striking contrast, PCDH19-CE affects heterozygous females, leaving males with hemizygous pathogenic variants apparently unaffected. Males generally do not present seizures or ID, though obsessive traits, subtle psychiatric symptoms or autism spectrum disorder (ASD) have been reported in some cases [1,6,28]. Of note, recent studies have identified males with ASD presenting pathogenic variants in PCDH19 [9,29]. While the peculiar expression of this disease is not compatible with negative dominance as the mechanism of disease, the postzygotic occurrence of pathogenic variants in symptomatic males supports “cellular interference” as a model to explain the genetic basis of PCDH19-CE [3]. This model postulates an unusual X-linked mode of inheritance in which females are more frequently affected than males. This model was adapted from the concept of metabolic interference proposed by Johnson in 1980 [30], which was used to explain the pattern of inheritance for cranio-frontonasal syndrome (CFNS), a disorder caused by mutations in *EFNB1*, encoding a protein, ephrin-B1, which is believed to be involved in cellular migration and adhesion [31]. In females, random inactivation of one X chromosome occurs in cells to achieve dosage compensation. In heterozygous females, random inactivation of one PCDH19 allele results in the coexistence of two cell populations expressing distinct PCDH19 proteins, which has been proposed to alter cell-cell interactions and proper neuronal functioning. Occurrence of a single copy of PCDH19 results in a single cell population in which the expression of the “pathogenic” allele does not seem to be dangerous for the function of the neuronal network. According to this theory, mosaic males are expected to be affected because of the coexistence of two different cell populations expressing different PCDH19 proteins. Indeed, the identification of mosaic males clinically expressing PCDH19-CE supports “cellular interference” as a reasonable model for this disease [3,28,32,33,34]. Of note, a non-mosaic male with Klinefelter syndrome (47, XXY) has also been reported to be affected by PCDH19-CE [35], thus further excluding the hypothesis of a compensatory rescue in males and confirming the “cellular interference” model.

Although PCDH19-CE can be either paternally or maternally inherited, pathogenic variants are mostly de novo [1,3,36]. Mutations in PCDH19 affect highly conserved amino acids, but have a relatively wide molecular spectrum, including missense and nonsense nucleotide substitutions, as well as small deletions or insertions. These mutations are all predicted to be inactivating [1,3]. To date, close to 200 PCDH19 pathogenic variants have been reported, which makes this gene one of the most commonly mutated in epilepsy [3,36]. Of note, high variability in clinical features has been reported for individuals carrying the same PCDH19 mutation [28,36]. No clinically relevant genotype-phenotype correlation has been reported in terms of seizure onset and disease severity; although, the age of onset of seizures appears to correlate with the overall severity of the phenotype [28]. The lack of a phenotype-genotype correlation may be explained by different patterns of chromosome X inactivation, which likely accounts for a large proportion of the observed phenotypic variability. At present, there is no correlation between the X-inactivation status of blood cells and symptoms [7], probably because the X-inactivation status of blood cells does not reflect the pattern occurring in the brain. In fact, the severity of the disease may be correlated with the amount of X-inactivation specifically present in a subpopulation of neurons in the brain.

## 3. PCDH19 in Neurodevelopment

PCDH19 is a membrane-bound calcium-dependent adhesion molecule belonging to the family of δ2-protocadherins [37]. These proteins consist of 6 tandemly arranged extracellular cadherin repeats (all encoded by exon 1), a single transmembrane domain, and an intracellular C-terminal tail with two conserved motifs (i.e., CM1 and CM2) [38]. Pathogenic variants of PCDH19 occur mainly in the first exon [1,28,36], suggesting that the extracellular domain is crucial for maintaining the protein’s functionality. In vertebrates, δ2-protocadherins are predominantly expressed in the developing and adult nervous system with specific spatial and temporal expression patterns [39], suggesting that they play important roles in neural development and function, such as neuronal differentiation, axon guidance [40], dendritic arborization [41] and self-avoidance [40], and synapse formation and dynamics [42,43,44]. The spatiotemporal expression, molecular diversity, and isoform-specific homophilic binding properties might provide a recognition code that supports neural circuit formation, proper neuronal connectivity, and synaptogenesis [45]. It has been demonstrated that some non-clustered PCDHs (i.e., PCDH10, PCDH17 and PCDH19), rather than stabilizing cell adhesion, mediate cell migration and cell sorting in vivo [17,21,38]. Each δ2-PCDHs subtype is expressed by distinct neuronal populations and has specific binding properties. Thus, each δ2-protocadherin, through its adhesive mechanism, exclusively mediates interactions between axons expressing the same δ2-PCDH subtype, and therefore, serves to support the sorting of axons that are derived from different groups of neurons during the formation of the neural circuit and its maintenance [39,45]. In zebrafish mutants, the loss of PCDH19 causes an increase in neuron production and a reduction in cell cohesion with impaired columnar organization of the optic tectum, resulting in defective visually guided behaviors [46]. As expected, homozygous zebrafish mutants did not exhibit gross defects in neural organization [46], which is in line with the human phenotype.

PCDHs can *cis*-interact with other PCDHs as well as with “classical” cadherins, increasing the specificity of their binding properties. Cells expressing a specific *cis*-complex fail to aggregate with cells that express a different combination of PCDHs [45]. As observed for other protocadherins, PCDH19 *cis*-interacts with “classical” cadherins. Studies on zebrafish demonstrated that PCDH19 forms a *cis*-complex with n-cadherin (ncad), which seems to act as a cofactor for enforcing the adhesive properties of PCDH19 and increasing the combinational diversity of adhesive specificity mediating morphogenic movements during brain development [47,48]. During zebrafish neurulation, loss of either PCDH19 or ncad disrupts convergent cell movements during anterior neural plate formation [47,49], while targeted silencing of PCDH19 results in ectopic folds and incomplete closure of the neural tube [49], suggesting there were alterations in the cell adhesion properties. Therefore, PCDH19 is necessary for proper neural tube formation and brain morphogenesis. The relevance of PCDH19-Ncad interactions was recently demonstrated by studies on heterozygous female mice with a mutated PCDH19, which documented hippocampal presynaptic dysfunction and cognitive impairments resulting from mismatched interactions [50].

Work with mouse models has shown that heterozygosity in PCDH19 correlates with the altered migration of cortical neurons [16,23,51] and abnormal cell sorting [17,21], suggesting impaired neurogenesis occurs in the cortical lesions of patients [14,17,18,52,53], though morphological analyses did not reveal any gross abnormalities in the brain structures of the mice [16,17,21]. Remarkably, the altered segregation of the two different populations of cells has been demonstrated in the developing brains of heterozygous PCDH19 KO female mice [17], but not in hemizygous PCDH19 KO male mice [54]. These results provide experimental evidence for the postulated “cellular interference” effect as a key pathogenic mechanism for PCDH19-CE.

An increase in neuron cell number was demonstrated both in zebrafish [46] and mice [22]. Of note, using an informative zebrafish model, Cooper and colleagues [46] demonstrated that neurons display reduced cell cohesion and are more dispersed and arborized. Silencing PCDH19 during mouse development, Lv et al. [51] reported that the neurons have an increased number of neurites and impaired synaptic connectivity. Moreover, analyses of the synaptogenesis in mosaic cultures of primary hippocampal neurons derived from WT and KO mice demonstrated a significant reduction in synaptic contacts, significantly altered neuronal morphology (increased neurite length and branching), and significantly higher neuronal network activity [55].

A murine model of PCDH19 loss of function showed reduced radial glia proliferation, while the rescue of PCDH19 function was found to suppress the differentiation of neuronal progenitors, suggesting that PCDH19 regulates neuronal progenitor fate, and that mutated PCDH19 promotes basal progenitor differentiation [20]. Neural progenitor cells (NPCs) from mice have been used to investigate the role of PCDH19 during neurogenesis and confirmed that a loss of function leads to the accelerated differentiation of NPCs [22].

## 4. iPSCs as a Model System for Studying PCDH19-CE

The study of neurodevelopmental disorders, such as PCDH19-CE, needs informative models for exploring the processes mediating neuronal development and function. Unfortunately, genetically modified animal models have failed to recapitulate critical aspects of the human disease, including the occurrence of seizures [56,57]. Previous studies with zebrafish and murine models (summarized in Table 1) have underlined a role for protocadherin 19 in early embryonic neurodevelopment, but these models have not been informative for the study of the cellular pathways implicated in the epileptic phenotype characterizing PCDH19-CE [16,17,21]. While a single study reported some behavioral defects in heterozygous animals, such as a decrease in the fear response and slight hyperactivity (particularly in response to stress) [21], other analyses focused on other behavioral/cognitive features (e.g., anxiety, social interaction, working and spatial memory) failed to detect any differences between heterozygous females and the controls [21]. Moreover, while cortical structural abnormalities have been identified in patients [14,15,16,17,18], no gross morphological alterations were found in the brains from mouse models [16,17,21]. Several animal models (i.e., zebrafish and murine models) have been used for the study of several neurodevelopmental pathologies (Table 1), as they rapidly recapitulate embryogenesis and offer the possibility of quickly characterizing the phenotype of interest in addition to representing a physiological multi-organ system. Despite these advantages, they may not faithfully recapitulate the clinical features and the severity of human PCDH19-CE (Figure 1).

In 2007, iPSCs were introduced as an alternative in vitro disease modeling system [24]. These pluripotent stem cells can differentiate, if properly instructed, into cortical neurons [25]. This technology has opened a new field for stem cell research by allowing the generation of stem cells without the use of embryos and the ethical problems associated with their use. One possible disadvantage of using the iPSC as a model is that the reprogramming efficiency remains low despite several advances that have been made with mRNA reprogramming in microfluidic systems [58,60] (Figure 1). Importantly, these cells are patient-specific and can recapitulate neuronal differentiation for investigating the molecular, morphological, and functional aspects of differentiated neuronal cells carrying patient-specific mutations, with the advantage that they have a genome that is identical to the donor patient. These properties make these cells an excellent model for studying the molecular mechanisms underlying rare genetic diseases. Compagnucci et al. [61] used iPSCs from healthy individuals to characterize PCDH19 function. In particular, the authors investigated the localization of PCDH19 in stem cells and during in vitro cortical neurogenesis, demonstrating that PCDH19 is localized at one pole of undifferentiated cells, suggesting a possible role for this protein for informing the position of one cell in relation to its neighbors. In addition, during cell division, PCDH19 is positioned at both poles of the mitotic spindle, suggesting an involvement in the orientation of the spindle and a possible role for controlling the type of division (symmetrical or asymmetrical). Following cortical neuronal differentiation, cells re-organize themselves from colonies into neural rosettes, which are composed of neural progenitors positioned with an internal lumen resembling the neural tube during in vivo neurogenesis. Interestingly, in the neural rosettes, PCDH19 is observed at the center of the structure, thus defining the proliferative zone and providing important positional information during neuronal development. In mature neurons, PCDH19 localizes in the plasma membrane at cell-cell contact sites [61].

A second study used human iPSCs derived from PCDH19 patients and confirmed that PCDH19 has a relevant function during neural rosette formation [22]. In particular, Homan and co-workers suggested that the coexistence of cells expressing mutated and wild-type PCDH19 proteins was associated with a loss of apical-basal polarity and an increased rate of neuronal differentiation. In a recent work [59], Borghi and co-workers confirmed accelerated cortical differentiation in vitro using iPSC derived from a mosaic male patient (PCDH19-iPSCs). As in [22], PCDH19-iPSCs were mixed with wild-type iPSCs to recreate the mosaic condition. Moreover, comparing the differentiation process of mixed iPSCs with control ones, it emerged that, as in the pathological condition, neural rosettes appeared earlier and showed a disorganized structure with a reduced lumen [59], in line with the findings by Homan and co-workers [22]. To investigate the mechanism underlying this early differentiation, the same authors showed that PCDH19 loss of function results in significant centrosome hyper-amplification in mitotic iPSCs and an increase in asymmetrical cell division by progenitor cells close to the center of the rosettes [59]. This altered cell behavior is most probably responsible for the increase in cell differentiation and defective neural progenitor cell proliferation [62]. Importantly, a recent work demonstrated that PCDH19 interacts with nedd1 (neural precursor cell expressed developmentally down-regulated protein 1), an important protein for spindle assembly during development [63]. These results suggest that PCDH19 has a central role in controlling cell division during human neurogenesis.

## 5. From iPSCs to Brain Organoids

Research with 2D cultures of neurons obtained from differentiating iPSCs has contributed to a better understanding of the molecular mechanisms of PCDH19-CE (Table 1); however, they cannot be used to investigate aspects related to neurogenesis and neuronal network organization. Moreover, iPSC-derived neurons can exhibit immature electrophysiological properties, and thus, this model is not ideal for studying the pathologies characterized by epileptic seizures [57]. These drawbacks might be solved by new frontiers in iPSC technology, such as organoid models [26]. This approach employs not only multiple cell types in vitro, it allows the organization of those cell populations into 3D structures with the same intrinsic patterning as the organ itself, resulting in an organization that closely resembles the endo-physiological architecture [26]. Thus, organoids represent a useful tool for investigating human organogenesis, particularly when considering the brain. In fact, pluripotent stem cells, cultured in 3D, are able to self-organize to form clusters of neuroepithelial cells with a latent intrinsic potential to produce stratified structures constituting layer-specific neurons generated in a sequential manner that imitates embryonic cortical neuroepithelium development [64,65,66]. Importantly, it has been demonstrated that in vitro neural differentiation of pluripotent stem cells is a cell autonomous process that recapitulates species-specific in vivo neurodevelopment [67,68]. Thus, iPSCs provide a valuable source of cells for understanding the mechanisms of in vivo human neurogenesis in the context of 3D brain organoids. These findings provide the basis for developing a methodology for obtaining a 3D in vitro model through the self-organization of iPSCs into “cerebral organoids”. These structures present proliferative zones that resemble the ventricular and the subventricular zones of the developing brain, with concentric cell layers expressing markers for different cortical layers and a cavity reminiscent of the brain ventricle [26]. Organoid-derived neurons are electrically functional in the presence of glutamate, since they show spontaneous Ca2+ peaks [26,27,69], and exhibit a spontaneous electrical activity that increases over time, thereby confirming that this technology can produce fully mature and functional neurons [70,71,72,73,74,75]. Furthermore, human organoids contain basal radial glial cells, which are lacking in murine models due to less expansion of the murine neural progenitor cells than their human counterparts, making this the methodology of choice for studying human brain development [76]. The generation of cerebral organoids makes it possible to model and investigate the pathophysiology of many neurological and rare disorders as well. Recently, brain organoids were used as a model system to investigate the neurodevelopmental processes in ASD [77], Miller-Dieker syndrome [78,79], Rett syndrome [80], Down syndrome [81], and in genetically determined macrocephaly [82,83] and microcephaly [26,84,85].

The first brain organoid model for PCDH19-CE was developed using iPSCs derived from a mosaic male patient [59]. The data showed that the patient-derived organoids were smaller than the controls [59], which is consistent with the mild microcephaly observed in some patients [14] and the increased neural differentiation observed in 2D cultures [59] and mouse models [20,22]. These findings suggest premature neurogenesis, similar to what has been observed for brain organoids modelling Miller-Dieker syndrome [78]. Interestingly, Miller-Dieker cerebral organoids showed mitotic defects in a subtype of neural progenitors, the outer radial glia [79], as well as cerebral organoids expressing a mutated ASPM (abnormal spindle protein, microcephaly-associated), a protein involved in progenitor mitotic spindle orientation [86], which displayed a smaller size and a reduced outer radial glia cell layer [84]. Brain organoids derived from iPSCs carrying a mutated *CDK5RAP2*, a centrosomal protein for which mutations perturb the formation of the mitotic spindle and have been linked to autosomal recessive primary microcephaly [87], also exhibited a smaller size due to altered mitotic spindle orientation in neural progenitor cells, which divided asymmetrically, resulting in a decreased pool of neural progenitors and premature neurogenesis [26].

Like these models, the smaller size of the organoids derived from PCDH19-iPSCs might also arise from a decrease in the pool of progenitors caused by abnormal mitotic division, which is supported by the observation of a mitotic spindle defect during 2D neural differentiation of PCDH19-iPSCs. The study of PCDH19-CE using brain organoids is just beginning, and it will be necessary to deepen our understanding of how PCDH19 influences the dynamics of neural progenitor division leading to impaired neurogenesis.

## 6. Blood-Brain Barrier and Brain Organoids

One disadvantage of brain organoids is related to the fact that this model approaches brain pathophysiology without considering a role for the vasculature. This issue is a great limit for two reasons: first, it seriously impairs the growth of 3D brain organoids into a considerable size due to the impaired availability of nutrients for the inner portions of these structures; second, they cannot be used to assess the ability of small molecules of interest to cross the blood brain barrier (BBB). In fact, in physiology, many drugs, despite their effectiveness on neural cells, cannot be used in the clinic as they do not cross the BBB. To understand the potential of an iPSC-derived BBB for performing pre-clinical screening of central nervous system (CNS)-targeting drugs, specific 2D human in vitro BBB models have been developed [88,89,90]. The BBB is a semipermeable barrier resulting from the presence of specialized tight junctions between the endothelial cells of the brain capillaries, which restricts the passage of toxins and pathogens, and regulates the movement of ions, molecules, and cells between the blood and the cerebrospinal fluid [91,92]. The function of the BBB is to preserve homeostasis in the brain microenvironment by preventing the passage of large or hydrophilic molecules (unless specific transporters are present on the luminal and abluminal membranes) while allowing the diffusion of hydrophobic molecules (i.e., O2, CO2, hormones) and small non-polar molecules [91]. Another role of the BBB is to protect the brain from the damage caused by peripheral immune events by limiting the passage of antibodies and immune cells into the CNS [93]. As a direct consequence of this role, the BBB prevents diffusion of many pharmacological agents, including some antiseizure drugs (i.e., phenytoin) [92,94,95,96], which is clinically relevant, as it becomes an obstacle for an effective therapeutic strategy for many neurological disorders.

For the successful treatment of CNS diseases, it is necessary to verify that the drugs can cross the BBB in adequate quantities to be clinically effective. To investigate the permeability of CNS therapeutics, a useful tool is to model the BBB in vitro. Recently, an innovative protocol has been developed for generating a 2D model of endothelial cells with BBB properties (i.e., well-organized tight junctions, appropriate expression of nutrient transporters, and polarized efflux transporter activity) from iPSCs, co-differentiating endothelial cells with neural cells [89]. An even more relevant tool for BBB modelling is the development of BBB organoids, obtained from the co-culture of endothelial cells, pericytes, and astrocytes [97]. These 2D, and more importantly 3D, models have the potential to be implemented into pre-clinical CNS drug discovery and pharmaceutical development processes. A combined iPSC-derived BBB and neuronal co-culture system could represent an option for quickly testing the efficacy of specific drugs for crossing the BBB and having an effect on the underlying neuronal phenotype [98].

In order to investigate the ability of a drug treatment to across the BBB, several methods have recently been developed using brain organoids complemented by a vascular system with the characteristics of the BBB [99,100]. Interestingly, Ahn et al. [101] recreated a vascularized brain in vitro, obtaining human cortical organoids penetrated by human blood vessels organoids (BVOs). The BVOs consisted of endothelial cells and mural cells enveloped by a basement membrane, resembling the cytoarchitecture of typical blood vessels. Importantly, they expressed the molecular markers for the BBB [101]. The integration of brain organoids with a BBB creates a vascularized in vitro system that is an optimal tool for investigating human cortical development and neurodevelopmental diseases, and to test the effectiveness of drug compounds on the neural cells after assessing their transport across the BBB. Despite these advantages, vascularized brain organoids cannot be used to evaluate drug toxicity in other organs as can be done with the in vivo models (i.e., zebrafish and murine models) (Figure 1).

## 7. Conclusions

Understanding of the physiopathology of PCDH19-CE is not an easy endeavor due to the inaccessibility of the tissue of interest, the difficulties of exploring morpho-functional alterations of the neural network in situ, and testing drugs to rescue the altered phenotype. Researchers have used two different animal models to study protocadherin 19 in a physiological environment, but with only partial success. On one hand, zebrafish do not apparently exhibit gross defects in neural organization [46]; on the other hand, Pcdh19 heterozygous female mice do not have spontaneous seizures [17] (Figure 1), morphological analyses did not reveal gross abnormalities in any of the brain structures [16,17,21], and no neuronal segregation has been observed in hemizygous KO male mice [54]. A model system recapitulating the human phenotype is therefore necessary to further our understanding of PCDH19-CE and suggest new therapeutic avenues. The in vitro human iPSC model has emerged as an informative model system that can account for the same genetic background as the patient and be used to develop 3D brain organoids with a BBB, thereby mimicking a cellular environment that is closer to human physiology (Figure 1). The development of 3D brain organoids from iPSCs has provided the first insights on PCDH19-CE and, by advancing the imaging of 3D organoids, they will certainly shed further light on the pathogenesis of this disease with an understanding of the electrophysiological details of neural network alterations. Brain organoids have been used to model many neurological diseases and, in the future, we envision that they can also be used to test novel pharmacological therapies.

## Figures and Tables

**Figure 1 ijms-23-03506-f001:**
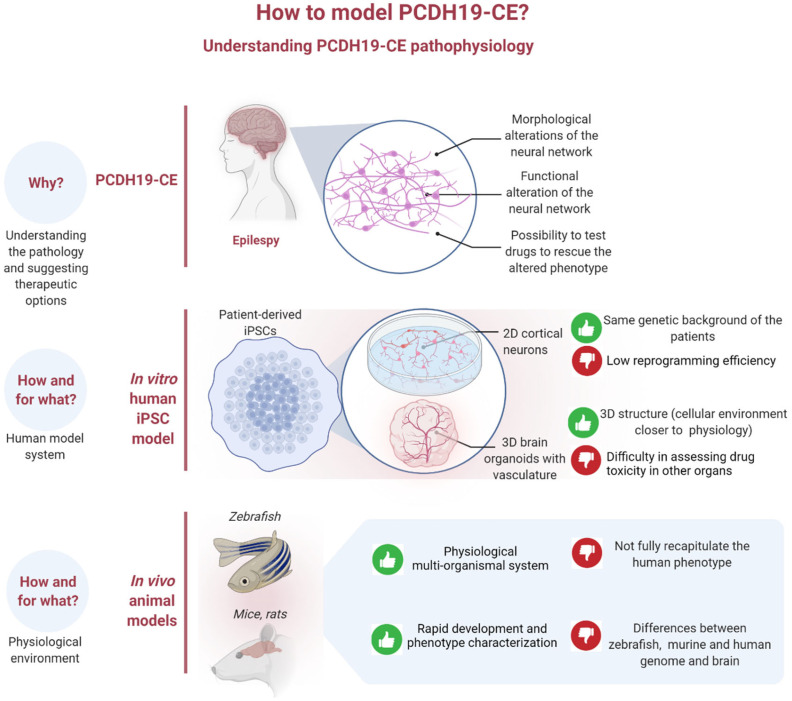
Schematic drawing depicting the available tools for understanding the pathophysiology of PCDH19-CE. The advantages of the specific models are indicated by a green thumb pointing upwards, while the disadvantages of the models are indicated by a red thumb pointing downwards. Created with BioRender.com.

**Table 1 ijms-23-03506-t001:** In vitro and in vivo models available for understanding the physiological and pathological role of PCDH19. This table summarizes our knowledge on the role of PCDH19, reporting the results from different in vitro and in vivo model systems.

**Understanding the role of PCDH19 using in vivo and in vitro models**	**Model**	**Reference**
Role in cell division (probable involvement in mitotic spindle orientation)	Human iPSCs	Compagnucci et al., 2015 [58];
PCDH19/PCDH19 localizes at the neuron-neuron cell contact sites	Human iPSCs	Compagnucci et al., 2015 [58]
	Mouse	Hayashi et al., 2017 [21]
pcdh19 is necessary for early stages of neurulation of the zebrafish embryo (disruption of convergent cell movements and impaired brain morphogenesis in PCDH19 KO embryos)	Zebrafish	Emond et al., 2009 [49]
pcdh19 cis-interacts with ncad, which acts as a cofactor to enforce the adhesive properties of PCDH19	Zebrafish	Biswas et al., 2010 [47]
	HEK293 cells	Emond et al.,2011 [48]
pcdh19 interacts with nedd1, an important protein for spindle assembly during development	HEK293 cells	Emond et al., 2021 [59]
**Understanding the PCDH19-dependent neurological alterations in in vivo and in vitro models**	**Model**	**Reference**
Mismatching between PCDH19 and Ncad interactions results in hippocampal presynaptic dysfunction and cognitive impairments	Mouse	Hoshina et al., 2021 [50]
Abnormal cell sorting and segregation of PCDH19^+^ and PCDH19^-^ cortical NPCs and their progeny in heterozygous mice	Mouse	Pederick et al., 2018 [17]; Hayashi et al., 2017 [21]
Impaired migration and altered localization of cortical neurons	Zebrafish	Cooper et al., 2015 [46]
	Mouse	Pederick et al., 2016 [16]; Lv et al., 2019 [51]
	Rat	Bassani et al., 2018 [23]
Altered neuronal morphology	Zebrafish	Cooper et al., 2015 [46]
	Rat	Bassani et al., 2018 [23]
	Mouse	Mincheva-Tasheva et al., 2021 [55]
Impaired synaptic connectivity of neurons	Mouse	Lv et al., 2019 [51]; Mincheva-Tasheva et al., 2021 [55]
Decreased fear response and slight hyperactivity	Mouse	Hayashi et al., 2017 [21]
Increased numbers of neurons	Zebrafish	Cooper et al., 2015 [46]
	Mouse	Homan et al., 2018 [22]
	Human iPSCs	Borghi et al., 2021 [60]
Accelerated neural differentiation	Mouse	Homan et al., 2018 [22];
	Human iPSCs	Borghi et al., 2021 [60]
Loss of apico-basal polarity of NPC	Human iPSCs	Homan et al., 2018 [22]
Reduced radial glia proliferation and increased radial glia differentiation	Mouse	Fujitani et al., 2017 [20]
Altered mitotic spindle and increased asymmetric cell division in progenitor cells	Human iPSCs	Borghi et al., 2021 [60]
Smaller size of patient-derived cerebral organoids compared to control ones	Human cerebral organoids	Borghi et al., 2021 [60]

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
