# Peer review of "Modeling PCDH19-CE: From 2D Stem Cell Model to 3D Brain Organoids"

_ijms, 2022, doi:10.3390/ijms23073506_

Round 1

Reviewer 1 Report

Dr. Borghi and colleagues reviewed about modeling methods about PCDH19. Generally, the content represents current understanding about this field. I suggest some modifications for more readability.

  1. In Figure 1, since rat is also used as animal model, it should be included in the Figure.
  2. In Figure 1, using arrows to indicated advantages and disadvantages will lead to confusion. Please modify them.
  3. In Figure 1, comparison of each model systems should include both advantages and disadvantages.

Author Response

Answers to Reviewer 1 

We thank Reviewer 1 for the appreciation of the manuscript and the usefuls comments and suggestions.

Comment 1: In figure 1, since rat is also used as animal model, it should be included in the figure.

Answer: As proposed we have modified Figure 1, including the rat as animal model in figure 1.

Comment 2: In figure 1, using arrows to indicate advantages and disadvantaes will lead to confusion. Please modify them.

Answer: As suggested by the reviewer, we have modified the arrows with more intuitive icons.

Comment 3: In figure 1, comparison of each model system should include both advantages and disadvantages.  

Answer: Following the reviewer’s suggestion, we have added the disadvantages of the iPSCs/organoid model and the advantages of the animal model.

Reviewer 2 Report

PCDH19 clustering epilepsy is a type of drug-resistant epilepsy caused by a mutation or partial deletion of the PCDH19 gene. This disease can cause harm such as intellectual disability, mental illness, and behavioral disorders. As animal models (i.e. zebrafish and mice) fail to fully characterize the phenotype and brain malformations of this disease in general, induced pluripotent stem cell (IPSC) technology provides a complementary experimental tool for studying PCDH19-CE. Therefore, the authors' study is relevant. In this review, the author first briefly introduces PCDH19-CE as a disease, discusses the current state of research on PCDH19-CE models, and presents the idea of using IPSCs technology to model the study of PCDH19-CE. Then, the author pointed out the correlation between PCDH19 and epileptogenesis in terms of genetics and neurodevelopment. Finally, the author presents the advantages of using IPSCs as a model system to study PCDH19-CE and provides insight into the leap from a two-dimensional stem cell model to a three-dimensional brain organ model in terms of brain organism and blood-brain barrier. However, this manuscript has some minor errors. My comments were shown below.

Major point 

1) The author pointed out the correlation between PCDH19 and epileptogenesis in terms of genetics and neurodevelopment. However, the feasibility and advantages of IPSCs as a model system for studying PCDH19-CE have been described in the latter section only in terms of neurodevelopment. Is there corresponding literature from a genetic point of view showing the great advantage and better prospect of using IPSCs as a model system to study PCDH19-CE?

2) In the manuscript, the author details the advantages of using IPSCs as a model system to study PCDH19-CE, from a two-dimensional stem cell model to a three-dimensional brain organ model. However, whether this technique currently has limitations and certain aspects of deficiencies, this is a guide for future research perspectives and directions.

3) In the manuscript, the author focuses on the advantages of using IPSCs as a model system to study PCDH19-CE. However, modeling using IPSCs to study the pathogenic mechanism of PCDH19-CE has been studied in depth by corresponding literature, and the author may consider adding relevant literature to further corroborate the advantages and prospects of IPSCs as a model.

4) The classification and analysis of manuscript form 1 are more detailed, however, only 1 or 2 articles are assigned to each subcategory, and the author may consider adding relevant literature.

Conclusion: It is suggested to make minor revisions and supplement relevant literature.

Author Response

Answers to Reviewer 2 

We thank the reviewer 2 for considering “the authors' study relevant” and for the comments and suggestions proposed.

Comment 1: The author pointed out the correlation between PCDH19 and epileptogenesis in terms of genetics and neurodevelopment. However, the feasibility and advantages of IPSCs as a model system for studying PCDH19-CE have been described in the latter section only in terms of neurodevelopment. Is there corresponding literature from a genetic point of view showing the great advantage and better prospect of using IPSCs as a model system to study PCDH19-CE?

Answer: We appreciate the reviewer’s comment, but in literature, the use of iPSCs as a model to study PCDH19-CE is limited to the two studies cited in the text: Homan et al. 2018 and Borghi et al. 2021. In the text (line 289) we cited studies where iPSCs were used to develop 3D model for genetic neurological diseases: “Generation of cerebral organoids makes possible to model and investigate the pathophysiology of many neurological and rare disorders as well. Recently brain organoids were used as a model system to investigate neurodevelopmental process in ASD [75], Miller-Dieker syndrome [76, 77], Rett syndrome [78], Down syndrome [79], and in genetically determined -type macrocephaly [80, 81] and microcephaly [26, 82, 83]."

Comment 2: In the manuscript, the author details the advantages of using IPSCs as a model system to study PCDH19-CE, from a two-dimensional stem cell model to a three-dimensional brain organ model. However, whether this technique currently has limitations and certain aspects of deficiencies, this is a guide for future research perspectives and directions.

Answer: We have added limitations to the 3D model of brain organoids in Figure 1.In addition to the changes to Figure 1, we amended the manuscript text by adding the following sentence (line 364): “Despite this, a limit of vascularized brain organoids is the impossibility of evaluating drug toxicity in other organs as in in vivo models (i.e. zebrafish and murine models) (Figure 1).”

Comment 3: In the manuscript, the author focuses on the advantages of using IPSCs as a model system to study PCDH19-CE. However, modeling using IPSCs to study the pathogenic mechanism of PCDH19-CE has been studied in depth by corresponding literature, and the author may consider adding relevant literature to further corroborate the advantages and prospects of IPSCs as a model.

Answer: We thank the reviewer for this comment, but the studies that use iPSCs as model to study PCDH19-CE are very few, and we paid attention to mention them all in the text (line 240): “In particular, Homan and co-workers suggested that coexistence of cells expressing mutated and wild-type PCDH19 proteins is associated with a loss of apical-basal polarity and an increased rate of neuronal differentiation. In a recent work [59], Borghi and co-workers confirmed the accelerated in vitro cortical differentiation using iPSC-derived from a mosaic male patient (PCDH19-iPSCs). As in [22], PCDH19-iPSCs were mixed with wild-type iPSCs to recreate the mosaic condition. Moreover, comparing the differentiation process of mixed iPSCs with control ones, it emerged that in the pathologic condition, the neural rosettes appeared earlier and showed a disorganized structure with a reduced lumen [59], in line with the findings by Homan and co-workers [22]. To investigate the mechanism underlying early differentiation, the same authors showed that PCDH19 loss of function results in a significant centrosome hyper-amplification in mitotic iPSCs and increase of asymmetric cell division in progenitor cells close to the center of the rosettes [59]. This altered cell behavior most probably is responsible for the increased cell differentiation and defective neural progenitor cell proliferation [60].”

In addition, to follow the reviewer’s comment we added two relevant references and the following sentence to the text (line 216): “One possible disadvantage in using the iPSC as a model is the low reprogramming efficiency, despite several advancements have been made with mRNA reprogramming in microfluidic systems [58, 59].”

Comment 4: The classification and analysis of manuscript form 1 are more detailed, however, only 1 or 2 articles are assigned to each subcategory, and the author may consider adding relevant literature.

Answer: We had some difficulties in understanding what Reviewer 2 meant in this comment and we interpreted "manuscript form 1" as a typo for “manuscript figure 1”. We have therefore modified the Figure 1 by removing the literature cited as a detailed reference list can be found in Table 1.

This manuscript is a resubmission of an earlier submission. The following is a list of the peer review reports and author responses from that submission.